# Causes of Sickness Absenteeism in Europe—Analysis from an Intercountry and Gender Perspective

**DOI:** 10.3390/ijerph182211823

**Published:** 2021-11-11

**Authors:** Elżbieta Antczak, Katarzyna M. Miszczyńska

**Affiliations:** 1Department of Spatial Econometrics, Faculty of Economics and Sociology, University of Lodz, 90-255 Lodz, Poland; 2Department of Public Finance, Faculty of Economics and Sociology, University of Lodz, 90-255 Lodz, Poland; katarzyna.miszczynska@uni.lodz.pl

**Keywords:** sickness absenteeism, gender inequalities, socioeconomic factors, Europe, regionality, geographically weighted regression

## Abstract

This study aims to extract and explain the territorially varied relation between socioeconomic factors and absence rate from work due to own illness or disability in European countries in the years 2006–2020. For this purpose, several causes were identified, depending on men and women. To explain the absenteeism and emphasize gender as well as intercountry differences, geographically weighted regression was applied. For men, there were five main variables that influenced sickness absence: body mass index, the average rating of satisfaction by job situation, employment in the manufacturing sector, social benefits by sickness/health care, and performing health-enhancing physical activity. For women, there were five main variables that increased the absence rate: the risk of poverty or social exclusion, long-standing illness or health problems, employment in the manufacturing sector, social protection benefits, and deaths due to pneumonia. Based on the conducted research, it was proven that the sickness absence observed in the analyzed countries was highly gender and spatially diverged. Understanding the multifactorial factors playing an important role in the occurrence of regional and gender-divergent sickness absence may be a good predictor of subsequent morbidity and mortality as well as be very useful to better prevent this outcome.

## 1. Introduction

Improving health around the world today is an important social objective that has obvious direct payoffs in terms of longer and better lives [1]. Those improvements influence the pace of income growth through many pathways, e.g., better health directly increases labor market participation and workers’ productivity [2,3], and increasing life expectancy creates incentives to invest in education, innovation, and physical capital [4]. Meanwhile, better health, particularly that of women, reduces fertility and spurs an economic transition from a state of stagnating income toward sustained income growth [5,6,7,8,9]. On the one hand, technological advancements in medicine and growing expectations of the population mean increasing demand for treatment. This, in most developed countries, has led to an increase in health spending. On the other hand, health, which together with education creates human capital, is simultaneously a key factor that determines economic growth [1,10,11]. According to human capital theory [12], human capital is used to generate gross domestic product (GDP). At the same time, sickness absence, which is the immanent result of a disease [13], directly leads to the under-utilization of an individual’s capital. It results in decreasing productivity and the creation of non-produced GDP. Health, as an essential component of human capital, supports workers’ productivity by enhancing physical capacity and mental capabilities [14].

Thus, disease causes a reduction in work resources and limited productivity. It may also result in disabilities or premature death, leading to two types of consequences. Firstly, the household income of the sick and their informal caregivers decreases. Secondly, enterprises employ less than one factor of production in the short term, which leads to other factors not being utilized—above all, capital. Then, the company’s production volume decreases. Of course, replacements and hiring new employees, if possible, over time lead to the initial production volume. However, it does not change the fact that the production that would have been made by the sick, had they not got sick, remains unprocessed. Thus, it represents a loss that serves as a means of measuring the cost (direct and indirect) of the disease for the economy and society [15,16,17,18]. The components of indirect costs include the loss of production caused by absenteeism, presenteeism, absenteeism and presenteeism of informal caregivers, premature death, and an inability to work [19]. The issue of structuring the effects of morbidity on selected diseases and its impact on the functioning of society and the economy, in the context of socioeconomic factors, is most often raised by analyzing sickness absence [20]. Apart from planned, desired absences (e.g., annual holidays) or unplanned absences, such as layoffs, sickness absence is the main category of employee absence. It is worth underlining that this distinction is not always easy to make. The first type of absences is not a big obstacle for the company because they are planned in advance, and their effects can be mitigated, for example, by rescheduling work priorities. However, annual leave (as guaranteed by law) is also sometimes used by employers as a tool to mask layoffs. This occurs when employees are encouraged to take holidays when the enterprise is struggling with financial difficulties [21]. 

As previously mentioned, planned, desired absences from work (e.g., annual holidays) can easily be “absorbed” by companies, as their effects can be planned for and mitigated; however, unplanned, undesired absences (e.g., illnesses, lay-offs) can disrupt production cycles and lead to material losses for both employers and employees. Lost productivity, labor turnover costs, and the disability burden have made sickness absence one of the top policy matters throughout Europe [22]. Although there is a large amount of evidence available on the different factors associated with sickness absenteeism, studies on transnational European trends are far more limited. International comparisons are thus urgently needed, since they may not only help to assess a country’s economic performance but also enable overall patterns across countries to be observed, thereby indicating which policies are working. This could be valuable for both public health and economic reasons. Additionally, research and intervention priorities can be identified from the examination of similarities and differences in sickness absence between countries. Finally, this work could indicate countries where important steps should be taken to help reduce sickness absence across Europe. We try, therefore, to fill the knowledge gap concerning disability, work absenteeism, and sickness benefits in selected European countries. This study thus aims to extract the transnational causes of sickness absenteeism in men and women separately. This approach was associated with significant differentiation in the value of the absence rate in terms of gender, which was confirmed in the literature on the subject [23,24,25]. Moreover, the following research questions have been posed: Is the disease spectrum different in the studied countries?Why and where are the sickness absence percentages higher for men than for women?Do social policy and sick-leave legislation systems influence sickness absence trends?Do work-related factors, such as job satisfaction, affect sickness absenteeism?

## 2. Theoretical Framework on the Sickness Absence Determinants

Generally speaking, the reason for an employee’s sickness absence is the incapacity to work caused by sickness [20] Thus, it might seem that only those factors relating to health determine the development of sickness absence. However, this is not entirely true. Sickness absence is a very complex and diverse phenomenon. It is linked with many different factors that determine the causes, frequency, and duration of sick leave. As Whitaker (2001) [26] emphasized, it is strongly influenced by factors other than health, while de Vries et al. (2018) underlined that sickness absence cannot always be equivalent to sickness itself [27]. Whitaker (2001) categorized the factors determining sickness absence into three levels: macro, organizational, and individual [26]. Meanwhile, Striker and Kusideł (2018) divided sickness absence into four basic groups: health factors and quality of living (e.g., living conditions, nutrition, wellbeing, lifestyle), demographic factors, economic factors (e.g., income, national economy, poverty), and the labor market situation (e.g., level of education, property sector, unemployment, place of work) [20]. Vuorio et al. (2019) systematized the factors that influence absenteeism into three different categories, with various sociodemographic-, health-, and work-related risk factors being shown to influence sickness absence [28]. Research on the development of sickness absence is also conducted in many ways. Some researchers focus on studying sickness absenteeism from the perspective of the costs borne by the employer and, at the same time, the economic and organizational consequences borne by the company. Another trend focuses on analyzing the development of sickness from the perspective of the employees’ age and education, reported disease entities, or gender. In addition, research was also carried out that considers the economic sector or cultural determinants. From a psychological point of view, the following factors have been investigated: stress [29,30], personality [31], and job satisfaction [32,33]. Keus Van De Poll et al. (2020) and OECD (2015) underlined that common mental disorders, such as depression or adjustment disorders, are among the main causes for sickness absence in many countries [34,35]. Depression-related problems also formed the basis of the study conducted by Casini et al. (2013) [25]. Slany et al. (2014) examined the relationship between psychosocial factors and sickness absence, and included factors concerning the labor market and company activity [23]. They underlined that sickness absence may be seen as a crucial indicator in occupational health studies, an opinion that was also shared by Whitaker (2001) [26], who conducted research in this field. Slany et al. (2014) also investigated occupational status [23], as did the authors in [24,26]. Several studies have explored age [28,36,37], gender [29,38], marital status [31,39], the presence of children [34], race, and ethnicity [40]. The impact of education on sickness absenteeism was investigated in [29,34,41]. The relationship between health- and lifestyle-related factors and absenteeism has been studied in [31,34,42]. These studies considered many determinants related to general health and sleep problems [31,42], mortality [37,43], BMI (body mass index) [42,44], self-related health status [37,43], or the COVID-19 pandemic [45,46]. One of the most frequently considered reasons for absenteeism was mental health problems and chronic diseases [25,28,37,41]. Another group of factors is related to the quality of life [47]. As part of these factors, the literature considered physical activity [44], job satisfaction [28,44], financial satisfaction [28], and social support [25], among others. Determinants in the field of social policy are also an important and not fully studied group of factors that influence sickness absenteeism; the issues related to it are discussed in [48,49,50].

The literature review presented above clearly indicates the complexity of the phenomenon and the open nature of the catalog of analyzed factors. However, most studies examined the determinants of sickness absenteeism in specific countries or based on models that do not distinguish individual economy, whereas little has been done so far to test this hypothesis on a cross-national and gender basis. This study is the first attempt at analyzing and extracting the causes of sickness absenteeism in each of the European countries. The study was based on employees from the 20–64 age group, separately for men and women. This approach was associated with significant differentiation of the value of the absence rate and was confirmed in the abovementioned literature on the subject. The inference was carried out using geographically weighted regression (GWR). By applying the GWR, we searched for the sickness absenteeism determinants (for women and men separately), which, with varying force and in a varying spectrum, affected the phenomena in Europe not only in the studied unit but also in neighboring countries during the study period. The findings indicate that geographical differences should be considered when investigating the empirical relationships between the selected factors and European sickness absenteeism of men and women. To the best of our knowledge, a study that considers such a broad approach has never been carried out so far. Therefore, the results of this study should be relevant when formulating strategic labor and healthcare policy recommendations. They could provide information for European governments, employers, and policy makers to design strategies that aid to reduce economic expenditures, increase workers satisfaction, assist employers, and eventually facilitate economic development of the country. Moreover, our findings suggest establishing changes in the European sickness absence legislation to make available comparable official data between countries. Finally, the data provide preliminary evidence of where to look for practices that can be implemented. 

## 3. Materials and Methods

### 3.1. Materials

As mentioned before, sickness absence is the basis of an employee’s absence from work. It is a problem not only from the employer’s perspective but also from the perspective of the state and lost GDP [51]. Thus, sickness absence, often called sickness absenteeism, is most often understood as the number of missed working days by employees due to diseases. It is also defined as the absence from work that is attributed to an employee’s sickness and accepted as such by the employer [26]. One way of measuring sickness absence is to calculate the sickness absence rate (1). According to Eurostat, it can be calculated as follows [52]:(1)$$\mathrm{sickness}\;\mathrm{absence}\;\mathrm{rate}=\frac{\mathrm{own}\;\mathrm{illness}\;\mathrm{or}\;\mathrm{disability}}{\mathrm{employed}\;\mathrm{people}\;\mathrm{aged}\;20–64\;}\cdot 100\%,$$

This sickness absence index (1) is considered as a global measure of health status and a marker of psychosocial and physical functioning for working populations.

We carried out the study of sickness absence rate among men and women for most of the European countries, i.e., Austria (AT), Belgium (BE), Bulgaria (BG), Croatia (HR), Cyprus (CY), the Czech Republic (CZ), Denmark (DK), Estonia (EE), Finland (FI), France (FR), Germany (DE), Greece (GR), Hungary (HU), Ireland (IE), Italy (IT), Latvia (LV), Lithuania (LT), Luxembourg (LU), Malta (MT), the Netherlands (NL), Poland (PL), Portugal (PT), Romania (RO), Slovakia (SK), Slovenia (SI), Spain (ES), Sweden (SE), the United Kingdom (GB), Lichtenstein (LI), Norway (NO), Switzerland (CH), and Turkey (TR). The analysis was conducted for the period 2006–2020 using openly available seasonally adjusted data obtained from Eurostat, the WHO, the OECD, and national statistical offices. The study was not performed merely for the European Union countries: in order to provide the most complete picture of the European sickness absenteeism, the perspective was extended to include Iceland, Lichtenstein, Norway, Great Britain, Turkey, and Switzerland. Thus, analyses were conducted for 32 European countries. As previously noted, the absence rate due to own illness or disability has been strongly related to gender and shows high relative regional variability (spatial non-stationary). Many variables, therefore, could be possible determinants of the phenomenon, both within and outside an individual’s control. Thus, we suggested a range of factors for the men’s and women’s absenteeism process defined in the literature. The data were classified into seven divisions and are presented Table 1.

The data set (Table 1) is not a complete panel because of the gaps in some variables during the analyzed period, with panel modeling being impossible for our data bank. We averaged the values of all the factors and conducted several stepwise regressions to identify the predictive variables of the phenomena over the time span. Finally, to overcome the panel-unbalanced, non-stationarity and spatial autocorrelation problems, we applied GWR to explain the absenteeism of men and women in Europe. 

### 3.2. Methods

Spatial data are more complex in their structure than in their time series. Analyzed spatial units are often diverse in terms of their size and economic or social potential, as confirmed by this study, where the database consisted of both small and large countries that differed significantly in their levels of absence rate for men and women (see Section 4.1). The relationships modeled in such cases often vary over geographical space, which causes heteroscedasticity of the random component in OLS (ordinary least squares) regressions. One of the ways to solve this problem is to use the GWR model, which considers spatial heterogeneity and dependency. This method makes it possible to identify the variability in regression coefficients in a geographical space by generating a separate regression equation for each observation (here for each analyzed country for men and women separately).

The GWR model can be expressed as follows [53]:(2)$$y_{i}=\beta_{0}\left(u_{i},v_{i}\right)+\sum \beta_{k}\left(u_{i},v_{i}\right)x_{ik}+\varepsilon_{i}$$
where *i* represents the spatial unit—here a country—*y_i_* is the dependent variable, (*u_i_*, *v_i_*) is the co-ordinate location of *i, β_0_* is the intercept, *β_k_* represents the coefficients, *x_ik_* represents the independent variables, and *ɛ_i_* is the error term.

The estimator for the model takes the form of
(3)$$\beta^{\prime }={\left[X^{T}W\left(u_{i},v_{i}\right)X\right]}^{-1}X^{T}W\left(u_{i},v_{i}\right)Y,$$
where W(*u_i_*,*v_i_*) is the square matrix of weights relative to the position of (*u_i_*, *v_i_*) in the study area, X*^T^*W(*u_i_*, *v_i_*)X is the geographically weighted variance–covariance matrix (the estimation requires its inverse to be obtained), and Y is the vector of the values for the dependent variable [54].

The W(*u_i_*, *v_i_*) matrix contains the geographical weights in its leading diagonal and 0 in its off-diagonal elements [55],
(4)$$W\left(u_{i},v_{i}\right)=\left[\begin{array}{ccc} w_{1}\left(u_{i},v_{i}\right) & 0 & 0\\ 0 & \ldots & 0\\ 0 & 0 & w_{n}\left(u_{i},v_{i}\right) \end{array}\right],$$
where *n* is the number of observations (spatial units).

Each equation was calibrated using different weights of observations contained in the dataset. According to Tobler’s first law of geography (“everything is related to everything else, but near things are more related than distant things”), which assumes that observations near one another have a greater influence on each other’s parameter estimates than observations farther apart, meaning that the impact decreases with distance [56], the weight assigned to each observation was based on a distance decay function centered around observation *i* [57]. The choice of the spatial weighting scheme, in particular, the kernel bandwidth, is one of the most important steps of the analysis because it determines the final results [53]. In this study, the adaptive type of the spatial kernel was used to provide geographic weighting in the model since the density of observations varied across the study area [55]. A key coefficient in the kernel is the bandwidth, which controls the size of the kernel. The corrected Akaike Information Criterion (ACIc) method was applied in this study to determine the bandwidth (the bandwidth for which the statistic takes the lowest values is considered to be optimal), as is recommended in the literature [54].

As with any GWR study, it is important to estimate the parameters of the global non-spatial regression (1) so that this benchmark model can be compared to its GWR counterpart [58,59]. However, as there is no single agreed-upon functional form in modeling, several statistical tests were conducted, using a pseudo-stepwise procedure, to explore the data with a limited number of OLS regression analyses [55]. To test for multicollinearity, the variance inflation factor (VIF) measure was used [60]. To test the spatial dependency on the residuals, Moran’s I and the Lagrange multiplier tests for both dependence error and missing spatially lagged dependent variable were used [61]. To identify the spatial non-stationarity, Koenker’s statistic (Koenker’s studentized Bruesch-Pagan test) was applied [62]. The local multicollinearity problem was subsequently solved and processed with the principal components method (PCA) [63].

The GWR generates localized parameter estimates and localized versions of all standard regression diagnostics, which can be mapped. Here, maps play a key role for interpreting the results and understanding the relationships from the regional and gender perspectives [59]. For estimations, we used ArcGIS software.

## 4. Results

### 4.1. Data Analysis

At the European level, the number of people who were absent from work due to their own illness or disability went up from 3.6 to 5.2 million between 2006 and 2020 (by 44%). This represents 2.7% and 2.9% of employed people, respectively, for these two specific years of the analyzed time span. However, the process unfolded at different rates for men and women. The rate of absences of women increased from 3.5% to 4.3% of employed people between 2006 and 2020. These numbers were lower for men and corresponded to 2.6% of employed people in 2006 and 3.1% in 2020, respectively. Between 2006 and 2020, the average annual pace of changes in women’s absence runs at about 1% from year to year, and it was 0.1 p.p. (percentage points) higher than for men (Figure 1). 

Between 2006 and 2020, there was 1 percentage point more women’s than men’s absences from work due to own illness or disability, and the diversification of the analyzed countries, measured with the coefficient of variation (CV), was high (CV = 159% for men’s and 151% for women’s absence). The Mann–Whitney U test revealed the statistically significant differences in the absences from work of men and women among countries, as can be seen in Table 2.

Germany stood out with the highest rate of absences among all countries (20.4% for men and 22.9% for women), followed by France (12.6% and 18.7%), Spain (8.8% and 11.9%), and the United Kingdom (7.9% vs 12.3%); Figure 2. Considering the data, the share of absences in employment was higher for women than for men in all countries. This is also observed at the average European level: 2.6% of employed men were absent from work against 3.6% of women (this statistically significant difference was also confirmed by the Mann–Whitney test; Table 2). The gender gap was the widest in France, where absences stood at 12.6% for men and 18.7% for women (gap 6.5 p.p., statistically significant at *p* ≤ 0.01), followed by United Kingdom (7.9% versus 12.3%, gap 4.4 p.p., statistically significant at *p* ≤ 0.01), Spain (8.8% and 11.9%, gap 3.1 p.p., statistically significant at *p* ≤ 0.05), and Poland (3.8% versus 6.3%, gap 2.5 p.p., statistically significant at *p* ≤ 0.05). The opposite tendency in the gender gap could be observed only in Turkey (2.1% for men versus 0.9% for women, gap −1.2 p.p., statistically significant at *p* ≤ 0.01) and in Romania (0.7% and 0.4%, gap −0.4 p.p., statistically significant at *p* ≤ 0.05). The narrowest gap was observed in Cyprus (0.08% versus 0.09%, gap 0.011 p.p. not statistically significant), followed by Estonia (0.2% versus 0.21%, gap 0.01 p.p., not statistically significant), Luxemburg (0.10% and 0.11%, gap 0.011 p.p., not statistically significant), and Iceland (0.05% versus 0.07%, gap 0.014 p.p., not statistically significant).

The European countries are characterized by large spatial distortions in absenteeism. On the other hand, countries can be grouped into some homogeneous areas. This was also underlined in the research of [66]. For a robustness check of these preliminary outcomes, we applied explanatory spatial data analysis (ESDA) tools—the global and local (local indicators of spatial association, LISA) spatial autocorrelation measures of Moran’s I, following Anselin and Florax (1995) [67] (Table A1 and Figure A1 in the Appendix). The global Moran’s I indices (Table A1) revealed that adjacent countries tended to cluster according to absences from work due to own illness or disability, but the LISA indices provided spatial clustering that remains stronger in terms of women’s rather than men’s absenteeism. In 2006 and 2020 in France (10.8% in 2006 and 15.4% in 2020) and Germany (10% in 2006 and 20.5% in 2020), there was the cluster with a higher proportion of men’s absence rate than women’s (in relation to countries which have a high number of cases). In Spain (13.5%), Great Britain (13.9% in 2006 and 10.4% in 2020), Germany (11.9%), and the Netherlands (10.4% in 2020), there was a higher number of absenteeism cases of women due to own illness or disability above the European average (3.5%) and in relation to countries which have a high number of cases. In turn, in Romania, Bulgaria, Lithuania, and Turkey, there was a cluster of countries with a low proportion of men’s absenteeism.

Further analysis of the GWR explores and explains why this spatial structuring is observed. It also occurred that sickness absence is associated with certain unobserved (hidden) regional patterns towards the spatial directional distribution of the process determinants (i.e., informal work, European labor law, the level of socioeconomic development, or the spread of diseases in the surrounding areas) [68,69].

### 4.2. Results of Modeling on Intercountry Sickness Absence

The preliminary data analysis showed that European absenteeism is gender-specific, and the differences regarding the absence rate of men and women should therefore be included in the modeling. Using ArcGIS software, we conducted several stepwise OLS regressions to identify the predictive variables of the absence rate of men and women in Europe between 2006 and 2020. Finally, to overcome all the problems (connected to regional variability, spatial non-stationarity and dependency, incomplete panel data set) and to model the phenomenon properly, we estimated the GWR functions for each gender category separately. We averaged the values of all variables and expressed them in natural logarithms as the log-log model better describes the relationship than other types of functions do. The regression results indicated the statistically significant relationship between the men’s (5) and women’s (6) absenteeism and various factors:(5)$$\begin{array}{c} AR_{men}{,}_{i}=\gamma_{0}\left(u_{i},v_{i}\right)+\gamma_{1}\left(u_{i},v_{i}\right)BMI_{i}+\gamma_{2}\left(u_{i},v_{i}\right)ARSJ_{i}\\ \;\;\;\;\;\;\;+\gamma_{3}\left(u_{i},v_{i}\right)MEM_{i}+\gamma_{4}\left(u_{i},v_{i}\right)SBSHC_{i}+\gamma_{5}\left(u_{i},v_{i}\right)PA_{i}+\varepsilon_{i}, \end{array}$$
(6)$$\begin{array}{c} AR_{women}{,}_{i}=\alpha_{0}\left(u_{i},v_{i}\right)+\alpha_{1}\left(u_{i},v_{i}\right)PRP_{i}+\alpha_{2}\left(u_{i},v_{i}\right)LSI_{i}+\alpha_{3}\left(u_{i},v_{i}\right)WEM_{i}\\ +\alpha_{4}\left(u_{i},v_{i}\right)SPB_{i}+\alpha_{5}\left(u_{i},v_{i}\right)CDP_{i}+\varepsilon_{i}, \end{array}$$
where $\left(u_{i},v_{i}\right)$ denotes the coordinates (longitude, latitude) of destination location *i,* for *i* = 1, 2,…, 32 countries, $\gamma_{k}\left(u_{i},v_{i}\right)$ are the structural parameters of the weighted regression model, and *ε_i_* is the random error at location *i* (the rest of variables are explained in Table 1). In the modeling, we selected statistically significant determinants with VIF values that did not exceed 2.5, and the local multicollinearity problem was subsequently solved with the principal components method (PCA) (the correlation matrix, and all results of the VIF and principal components analysis are available on request).

The application of GWR functions (5) and (6) significantly improved the results of the modeling. All measures indicated that the GWR models had a markedly better fit to the empirical data (Table 3). The AICc value for men declined from 141.4 in the global model to 103.5 in the GWR; for women, it ranged from 107.1 in OLS to 58.7 in GWR. The value of the R-squared improved as well. For men, it increased from 0.41 in OLS to 0.66 (the average value of the adjusted local R-squared) in the GWR; for women, it rose from 0.48 to 0.75. The Jarque–Bera statistic indicated that the residuals were normally distributed, and importantly, the Moran’s I test was not statistically significant for either the women’s or the men’s GWR functions. Therefore, there was no spatial autocorrelation in the residuals (Table 3).

The local values of the coefficients in the GWR models are presented in Table 4.

There was a positive and statistically significant correlation between the men’s body mass index (*BMI_i_*) and the absence rate from work due to own illness or disability in approximately 40% of the analyzed countries. The highest values of the coefficients (from 5.7% to 7.3%) were observed for men in countries located in southern Europe, e.g., in Malta, Italy, Spain, Portugal, Greece, Cyprus, Turkey, and Switzerland, and in the western parts of Europe, e.g., in Iceland, France, and Ireland. The lowest significant impact was observed for men in Romania (the absence rate rose to 5.1% due to a 1% increase in the BMI index). A negative correlation between men’s average job satisfaction rating (*ARSJ_i_*) and absenteeism from work due to own illness or disability was noticed only in 8 out of the 32 analyzed countries, namely, in Portugal, Spain, France, Denmark, Ireland, the United Kingdom, Netherlands, and Iceland (a decrease in absenteeism from 11.9% to 23.1%). The strongest relationship was noticed for Portugal, but the highest for Great Britain and Iceland (−23.1% and −18.3%, respectively). Regarding the situation in the labor market (the share of men employed in the manufacturing sector in %, *MEM_i_*), the relationship with the dependent variable was positive and spread over most countries (in 88% units from all). Nonetheless, the highest parameter values (a rise in absence rate from 3.6% to 3.9%) were recorded in Germany, Belgium, Switzerland, the Czech Republic, and Luxemburg. The lowest values (an increase from 1.9% to 2.5%) were noted in Lithuania, Poland, Hungary, and Norway. A statistically insignificant relationship was observed only in Sweden, Finland, Latvia, and Estonia. The increase in social benefits by sickness/health care in PPS per male inhabitant (*SBSHC_i_*) also has a positive impact on the absence rate in 26 (80%) of the analyzed countries. An increase of 1% of this variable resulted in the highest rise (from 2.2% to 2.6%) of men’s absenteeism in Italy, the Czech Republic, Malta, Iceland, Austria, and Switzerland (ceteris paribus). The lowest values of coefficients were found in eastern and north-eastern Europe (from 1.6% to 1.9%). The last statistically significant variable was the share of men performing health-enhancing physical activity (*PA_i_*). The negative correlation between the absence rate and this factor was observed in the south-eastern and eastern parts of Europe—in Poland, the Czech Republic, Slovakia, Austria, Slovenia, Croatia, Italy, Romania, Bulgaria, Greece, Malta, Cyprus, and Turkey. The highest coefficient values were noted in the Czech Republic, Slovenia, and Austria (a decrease in absences from 1.3% to 1.7%).

A positive correlation between the share of women at risk of poverty or social exclusion in the total population (*PRP_i_*) and their absences from work was observed in 11 of the analyzed countries: Iceland, Norway, Ireland, France, Portugal, Spain, Italy, Greece, Turkey, Malta, and Cyprus. The strongest relationship occurred for women in Iceland, Ireland, and Malta (a rise in absenteeism from 3.6% to 4.9%, ceteris paribus). In the other countries, a 1% increase in the share of women at risk of poverty or social exclusion in the total population led to an increase in the absence rate by 3.1% to 3.5%. A positive relationship between absences and having a long-standing illness or health problem (*LSI_i_*) was noticed in 15 countries located in the north-western and eastern Europe. This factor has the highest impact on the absence rate in Great Britain, Iceland, France, and Norway. An increase of 1% in the share of women with a long-standing illness or health problem generated an average increase in the absence rate from around 2.2% in France to as much as 3.2% in Great Britain (ceteris paribus). The share of women employed in the manufacturing sector factor (*WEM_i_*) had a statistically significant and positive impact on women’s absence rate in most of the analyzed countries, apart from Sweden, Finland, Estonia, Luxemburg, Denmark, Belgium, Latvia, Czech Republic, Slovakia, Slovenia, Hungary, and Croatia. The impact was the highest for Italy, Bulgaria, Turkey, Malta, Greece, Romania, Cyprus, and Iceland. A 1% increase in women’s employment in the manufacturing sector generated an average rise in the absence rate from work due to illness or disability from 2.2% to 2.7%, ceteris paribus (however, still higher parameter values were recorded for men than for women; Table 4). In terms of its regional range, the social protection benefits in PPS per female inhabitant (*SPB_i_*) was the factor that most considerably affected the absence rate from work (this impact was noted in 72% of countries). An increase of 1% in social protection benefits generated an average increase in women’s absences from around 2.6% in Latvia, France, and Austria, to as much as 4.3% in Iceland (ceteris paribus). The last important factor affecting the women’s absence rate from work due to own illness or disability was deaths due to pneumonia per 100,000 inhabitants (*CDP_i_*). A positive and statistically significant relationship between these two variables was noticed in 34% of the analyzed countries: Iceland, Switzerland, Great Britain, Ireland, Iceland, the Netherlands, Belgium, Germany, Denmark, Cyprus, Lithuania, and Poland. The highest impact was observed for women in Belgium, Germany, the Netherlands, and Poland (from 1.7% to 2.1%, ceteris paribus).

## 5. Discussion

For the period analyzed, in the case of men’s absenteeism from work due to illness or disability, there were five main variables that influenced the rate of absence. An increase in body mass index (BMI) and employment in the manufacturing sector were associated with a rise in male sick leave, while an average rating in relation to job satisfaction, the social benefits available thorough sickness/health care, and engagement in health-enhancing physical activity reduced the absence rate for men in the selected European countries. Regarding women’s absenteeism from work due to illness or disability in the years between 2006 and 2020, there were five main variables that increased the absence rate: the risk of poverty or social exclusion, long-standing illness or health problems, employment in the manufacturing sector, social protection benefits, and cases of pneumonia culminating in death.

Risk factors related to lifestyle, such as smoking, alcohol use, high BMI, and low physical activity in working populations, accounted for a substantial proportion of the years of work lost due to disability and premature mortality. The figures obtained relating to body mass index and the absence rate from work due to personal illness or disability are in line with international research. Harvey et al. (2010) underlined that people with obesity take an extra four days of sick leave annually compared with employees of a normal weight [70]. However, in Eastern Europe and the Mediterranean, compared to the countries of Western and Northern Europe, higher rates of obesity in men are observed [71]. In many countries, this can be linked to the massive investment made by the food industry in advertising foods rich in sugar, fats, and preservatives, which may work to promote obesogenic behaviors in adults. For example, it has been suggested that a high degree of market liberalism has helped increase the prevalence of obesity in the Icelandic male population [72]. In Ireland, Malta, Iceland, Greece, Bulgaria, and Cyprus, where the highest BMI for men in Europe was observed (equal to or more than 25) [73,74], obesity among working adults was associated with a significant increase in male absenteeism [75,76]. Moreover, recent studies have reported both male patients and physicians in Portugal, Italy, and Spain to have a poor understanding of obesity [77,78,79]. In Turkey, the higher level of obesity among adult men compared to women is associated with quitting smoking, alcohol consumption, and occupation (the highest prevalence being among tradesman, officers, and the retired) [80]. Studies on the risk of male sickness absence from work in France found a high or moderate rating for both smoking and obesity in sickness absence due to circulatory diseases, and an association of low physical activity with sickness absence due to respiratory diseases [81].

On the other hand, in this study, we found a higher prevalence among women than men for sickness absence due to a long-standing illness or health problem. Our results are in line with the findings of [30,82,83], who also tested and verified the association between chronic diseases and sickness absence. However, results showed that some countries (Norway, Germany, the Netherlands, Luxemburg, Poland, the Czech Republic, Austria, and Slovenia) had the highest prevalence of long sickness absence, and some others had the lowest prevalence (Greece, Spain, Ireland, Turkey, and Romania), for both genders. These differences are partly attributable to political regimes, social security systems, sick-pay policies, and sick-leave legislation systems (e.g., the minimum contribution or employment period required to receive sickness benefits, sickness insurance processes, sickness certification rules), which vary substantially across Europe [84]. Work absences due to long-standing sickness generate substantial costs for social protection systems [15,16,48]. Our modelling outcomes indicated that the propensity for men and women to take days off work is significantly higher in countries where employees are entitled to full salary during sickness. For example, in Malta, Austria, and Italy, employers are required to continue paying full wages when employees fall sick, whereas in Greece, Ireland, Turkey, and Great Britain, these benefits take the form of a lump-sum allowance. In Denmark, the level of benefit received depends on the individual’s rate of pay and the number of hours worked [84]. In contrast, sick-leave compensation remains the responsibility of employers for much longer in the Netherlands and Switzerland (in both cases generally up to two years), and many employers choose to take out insurance to cover this [85]. Our findings on sickness absence in relation to country and gender differences show that in Norway, Sweden, Iceland, Malta, the Netherlands, and Denmark, women were more often absent from work due to long-standing sickness than their male colleagues. In Sweden, women took twice as many sickness absence days as men. Paid sick leave, therefore, is concentrated on women. One might expect that the propensity of employees to be absent from work would be higher in countries where the conditions of access to sickness benefits are the least restrictive and where entitlements are the most generous [48]. However, when analyzing these figures, national labor market structures, and employment rates by gender and age, need to be considered. Intercountry data on male and female sickness absence and labor supply suggest that the gender gap in sickness absence is also associated with the increased participation of women in the labor force (e.g., in Sweden and Norway compared to France) [86]. In Switzerland, the Czech Republic, Germany, Iceland, and Italy, men are much more likely to be a part of the labor force than women [87,88,89].

Unlike previous studies that analyzed similar questions regarding the reverse correlation between the average rate of job satisfaction and absenteeism from work due to personal illness or disability [14,32], we observed that the strength of that relationship was gender- and country-dependent. Our investigations showed the predicted impact of job satisfaction on men’s absences from work, especially in Portugal, Great Britain, Iceland, Spain, Denmark, Ireland, and the Netherlands. Generally, job satisfaction (negatively associated with absence) was found to be the most significant predictor of absenteeism, i.e., the more satisfied employees are with their workplace, the less likely they are to be absent, and conversely, the more dissatisfied employees are with their workplace, the more likely they are to be absent. [90]. However, the job satisfaction–absenteeism relationship, as it connects to personal feelings or beliefs associated with work, varies widely within organizations and between different sectors [32]. As we confirmed in this study, though, it is also country- and gender-specific. For example, the rising level of job satisfaction among Portuguese men reduced the rate of absenteeism from work and can be connected to the relatively high employment rate for women. Furthermore, the relatively high proportion of women working full time needs to be explained in the light of the poor Portuguese wage level. This nexus presumably connects to a lack of job satisfaction among women arising from their disadvantaged position in this country [91]. On the other hand, the relevant literature shows that British women tend to be both more dissatisfied with their job and more frequently absent than men, most of the time for unexplained reasons. This gap may be due to differences in the social roles that women and men play as a result of differential socialization. British women are, it seems, more frequently absent than men due to domestic and general health issues [32]. Danish and Irish male workers declared a substantially higher level of satisfaction across all job domains than workers in Mediterranean countries. Overall job satisfaction in Denmark and the Netherlands was due to the lack of a wage effect on satisfaction in job domains other than pay; a compensating negative wage effect on distance to job satisfaction in these two countries; and the effect of pay satisfaction on overall job satisfaction [92]. Finally, empirical research on job satisfaction and its relation to gender inequality evidenced the paradox that a gender job satisfaction gap does not appear in countries that have higher levels of gender labor market equality, such as Denmark, Finland, and the Netherlands, or in Portugal, where men enjoy better working positions and have higher job satisfaction than women [93].

In the case of employment in the manufacturing sector, the relationship with the dependent variable was regionally divergent but positive in nearly all the analyzed countries, both for men and women. One of the major explanations for gender differences in European work absenteeism, as displayed in medically certified sick leave, is the different roles that men and women occupy in society, in both the private and the professional domain. Given the vast range of activities and production techniques covered by the manufacturing sector, with the work involving activities such as construction, the physical or chemical transformation of materials, substances, or components into new products, and the production of goods using machinery, it seems rather obvious that manufacturing services should account for many of the jobs held by men [94]. Moreover, because of the tradable nature of manufactured goods (export and import), specializations and the business economy in this sector vary greatly between countries in Europe. Among the five largest European states, Germany and the Czech Republic stood out for having manufacturing sectors that contributed to one third of their regional employment and value added [95]. Moreover, due to the high industrialization of some areas in the country, in Germany (followed by France and Belgium), it is the manufacturing sector that records the highest level of accidents among men [96]. On the other hand, a large body of empirical research showed that shift work was also significantly associated with absence due to occupational accidents, primarily in the manufacturing sector [97,98]. For example, according to the Labor Force Survey, non-standard working hours, such as shift work, are more frequent in Belgium (especially in the manufacturing and health care sectors) than in Europe as a whole [99]. The overrepresentation of male workers in Belgium working unusual and flexible hours can be explained by Belgian machinery factories applying shift work practices (plant and machine operators work in industrial factories) [100]. However, in the industrial northwest of Italy, for example, the female workforce was more involved in the manufacturing sector than elsewhere [101]. In Italy, Greece, Austria, Turkey, and Spain, more female migrants took low-quality jobs compared with migrants in the northern part of Europe. Moreover, women, especially ethnic and migrant women, were overrepresented in these countries in the seasonal work sector—including tourism and agriculture, working in family businesses, leather manufacturing, retail, tutoring, harvesting, and in many cases jobs in catering [102,103,104,105,106,107].

Our findings confirm the assumption that increasing levels of physical activity are likely to influence work-related outcomes, such as reducing male sickness absence [108]. However, we found heterogeneity across European countries in relation to the absence rate among men and indications of an active lifestyle. In general, the prevalence of physical inactivity in Europe was higher for women than men during the period analyzed [109]. Countries across Europe apply different strategies for promoting participation in sport and physical education [110,111]. Denmark, Germany, Austria, and Slovenia appear to be more successful in promoting physical activity compared to other parts of Europe. One explanation might be that these European countries adopt more effective physical activity programs, with these initiatives being characterized by a higher number of participants [112,113]. On the other hand, the Czech Republic is ranked among the more active countries, as reflected in its high proportion of active adults (47%) and its higher levels of physical activity in men compared with women. The Czech Republic can also be considered as a country of walkers and cyclists [114].

On the issue of absence from work, we did also find an association between the proportion of women at risk of poverty or social exclusion and the rise in sick-leave rate in some countries, primarily Norway, Iceland, Ireland, Malta, and Cyprus. The risk of poverty is far higher in certain countries, particularly among women who live alone (ranging from 22% in Greece and the Netherlands to 81% in Norway and Iceland) [115]. In general, women in countries (Southern and Central Europe) with traditional family policy models are more likely to report poorer health than men. Living in conditions below the poverty line may bring a higher exposure to agents that cause or exacerbate certain illnesses (e.g., asthma), because of the poorer living and/or working circumstances [116]. Despite increases in their participation in the labor force in European countries, women continue to be engaged in the workforce less than men. They are also more involved in unpaid work, work in jobs that tend to be more precarious, are underrepresented in senior management and decision-making positions, earn less than men, and are more likely to end their lives in poverty. These findings suggest that increases in poverty will have detrimental effects on the health of individuals and confirm and refine the results obtained in the literature [117].

Finally, we found a statistically significant association between the women’s absence rate from work due to own illness or disability and deaths due to pneumonia. The highest impact was observed in Belgium, Germany, the Netherlands, and Poland. Pneumonia mortality has been increasing in half of the studied countries, but Poland, Germany, Austria, Czechia, Slovakia, Great Britain, and Hungary recorded the most deaths from pneumonia [118]. In general, death rates among women was higher throughout the observation period, typically in Eastern Europe [119]. These could be linked to the rise in the uptake of smoking among women over successive generations [120]. Moreover, for European men, smoking-attributable mortality is already declining, whereas for women, smoking-attributable mortality is still increasing, except in selected north-western European countries [121]. Chronic obstructive pulmonary disease may result also from longstanding exposure to tobacco smoking, occupational chemical substances, and indoor and outdoor air pollution, with a role played by genetic susceptibility, poverty, stunting, and bronchial infections, such as tuberculosis [122]. However, more research may be needed on the contribution of these factors to women’s sickness absence rate and deaths due to pneumonia.

## 6. Conclusions

Sickness absence is perceived in an ambiguous and multifaceted manner, and it can be viewed from different points of view. Sick leave is a burden for the employer and causes loss of productivity. Each disease generates different types of costs, including direct, indirect, and social costs. The magnitude of indirect costs results from, among other things, the loss of productivity associated with prematurely leaving the labor market, absence from work, and ineffective presence at work, which has also become the subject of research and analysis, not only in economics, but also in medicine. The literature on the subject considers various factors that determine the development of sickness absenteeism. Research also points to different categories of these factors. A better understanding of the absence from work due to illness is a great concern in many countries, emphasizing the importance of knowing the modifiable risk factor targets for disability prevention, maintaining an active work force, improving comprehension of the mechanisms behind the gender gap in sickness, and extending working lives. From a practical point of view, the results of our research may also contribute to the understanding of the broader issue of gender differences in illness behavior and may be informative for policies and interventions aimed at safeguarding gender equality in the labor market and reducing sickness absence. Therefore, the aim of the study was to extract these factors. The study indicated that sickness absenteeism is different in the analyzed countries and that the social policy has an impact on sickness absence, as does job satisfaction, but only in terms of men.

Among all the analyzed factors, the share of men performing health-enhancing physical activity had a statistically significant impact on reducing the absence rate of men. The average growth in the rating of satisfaction regarding job situation had the greatest impact. An increase in sickness absence is also influenced by an increase in BMI, the share of men employed in the manufacturing sector, and social benefits by sickness/health care in purchasing power standard per male inhabitant. However, the last two factors show the greatest spatial diversity of the impact. The countries of southern and south-eastern Europe are most affected by the increase in sickness absence among men. A common factor that increases the absenteeism of women and men is the share of men and women employed in the manufacturing sector. However, given the strength and spatial extent, this factor has a greater and more extensive impact on raising men’s absence rate. Generally, in most Western units, women account for a large part of sickness absence from work. The increase in women’s sickness absence is determined to the greatest extent by the share of women at risk of poverty or social exclusion in the total population and the social protection benefits in PPS per female inhabitant. The weakest impact was shown by deaths due to pneumonia per 100,000 inhabitants. The increase in social protection benefits in PPS per female inhabitant was the strongest, both in terms of the value of the factor and the spatial extent. The country most burdened by the increase in women’s sickness absence was Iceland and the countries of southern and central Europe.

Moreover, knowledge about factors that may cause sickness absence might prove useful for reducing sickness absence rates for both men and women in European countries. However, among the analyzed determinants, no factor was identified that would reduce absenteeism due to disease in women. This is one of the many reasons for further research in this area. Perhaps factors relating to having children, caring for them, state support for single mothers, or factors that influence the taking of sick leave during pregnancy should be analyzed. Another important issue from the point of view of women is sickness absenteeism related to a child’s disease, and it should be emphasized that it is a bigger problem for women than men. The cultural and work culture aspect, which was not developed in this study, seems to be extremely important, and the authors consider extending future research to include this aspect. Finally, the results obtained in this study may provide a better understanding of the potential sources and mechanisms behind gender differences in sickness absence and could therefore be informative for policies and interventions aimed at safeguarding gender equality in the labor market and reducing sickness absence, as well as for physicians who certify sick leave.

Although we did not find a statistically significant relationship between the effects of the COVID-19 pandemic on the sickness absence rates for both men and women, some recent studies provide evidence that women, particularly those with young children, are more often absence from work than men on average. Therefore, further research on the absence rate in general and the effects of COVID-19 on the labor market is warranted.

## Figures and Tables

**Figure 1 ijerph-18-11823-f001:**
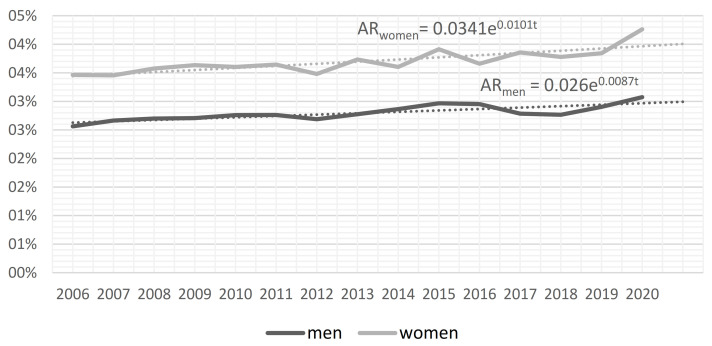
Dynamics of the absence rate for men and women due to own illness or disability in European countries, 2006–2020. Note: AR—sickness absence rate; t—time.

**Figure 2 ijerph-18-11823-f002:**
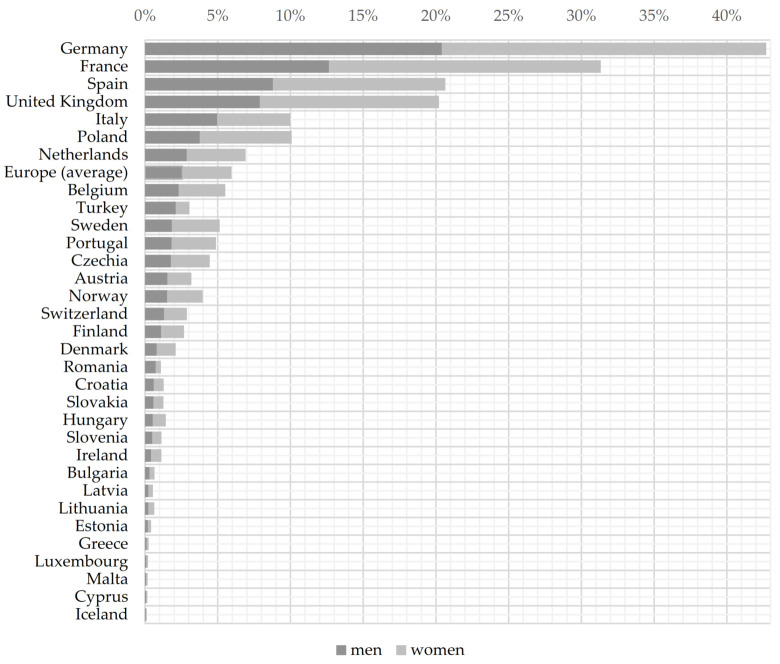
Average absences from work due to own illness or disability in Europe, 2006–2020, by sex, as % of employment. Note: *n* = 32 countries. To confirm the statistically significant difference of the gender gap, we used nonparametric (Mann–Whitney U) or parametric (Student’s *t*-tests) statistics, depending on the results of the Shapiro–Wilk test of normality; The results are available from the author on request.

**Table 1 ijerph-18-11823-t001:** Potential determinants of absenteeism in men and women.

Variable	Time Span	M—Men
		W—Women
		T—Total
HEALTH CONDITION		
In-patient average length of stay. Pregnancy, childbirth, and puerperium (O00–O99) (days)	2006–2018	W
People with a long-standing illness or health problem. Employed people except for employees (%)	2009–2020	M, W
Accidents at work (4 days or more) (standardized incidence rate)	2008–2018	M, W
COVID-19 (cases per 100,000)	2019–2020	T
Deaths due to diseases of the circulatory system per 100,000	2011–2019	M, W
Deaths due to pneumonia per 100,000	2011–2019	M, W
Self-perceived health. Bad or very bad (%)	2008–2020	M, W
Body mass index (BMI)	2014	M, W
QUALITY OF LIFE		
Performing health-enhancing physical activity. Aerobic and muscle-strengthening (%)	2014	M, W
Daily consumption of fruit and vegetables—5 portions (%)	2014	M, W
Overall perceived social support (%)	2014	M, W
Self-reported unmet needs for medical examination (%)	2008–2020	M, W
Overall perceived social support (%)	2014	M, W
Average number of rooms (per person)	2006–2020	T
Overcrowding rate (%)	2006–2020	M, W
Percentage of total population reporting exposure to pollution, grime, or other environmental problems (%)	2006–2020	M, W
Frequency of being happy in the last 4 weeks. Always and most of the time (%)	2013, 2018	M, W
Frequency of participation in cultural activities in the last 12 months. Cultural activities (cinema, live performances, or cultural sites) (%)	2006, 2015	M, W
Frequency of participation in sports activities (sports events) in the last 12 months (%)	2006, 2015	M, W
Intentional homicide victims (per hundred thousand inhabitants)	2008–2018	M, W
Average rating of satisfaction by financial situation (rating 0–10)	2013, 2015	M, W
Average rating of satisfaction by job situation (rating 0–10)	2013, 2015	M, W
ECONOMICS		
Distribution of households by household type–single person with dependent children (%)	2011–2020	T
Mean equivalized net income (purchasing power standard (PPS) per capita)	2006–2020	M, W
Disposable income and net lending (constant PPS per capita)	2006–2019	T
DEMOGRAPHY		
People at risk of poverty or social exclusion (%)	2011–2020	M, W
Proportion of population aged 15–24 years (%)	2009–2020	T
Proportion of population aged 25–49 years (%)	2009–2020	T
Proportion of population aged 50–64 years (%)	2009–2020	T
Proportion of population aged 65–79 years (%)	2009–2020	T
Old-age dependency ratio 3rd variant (population 65 and over to population 20 to 64 years) (%)	2006–2020	T
Women (per 100 men)	2006–2020	W
Median age of population (years)	2006–2020	M, W
EDUCATION		
Population by educational attainment level. Less than primary, primary, and lower secondary education (levels 0–2) (%)	2006–2019	M, W
Population by educational attainment level. Tertiary education (levels 5–8) (%)	2006–2019	M, W
LABOR MARKET		
Duration of working life (years)	2006–2019	M, W
Employed people with a second job (% of employed in group of 20 to 64 years)	2008–2019	M, W
Labor force participation rate (%)	2006–2019	M, W
Unemployment by sex and age (% of active population)	2006–2019	M, W
Employment in agriculture, forestry, and fishing (%)	2008–2019	M, W
Employment in mining and quarrying (%)	2008–2019	M, W
Employment in manufacturing (%)	2008–2019	M, W
Employment in electricity, gas, steam and air conditioning supply (%)	2008–2019	M, W
SOCIAL POLICY		
Wage replacement (%)	2009–2020	T
Employer responsibility for funding (length in days)	2006–2020	T
Social protection benefits (purchasing power standard (PPS) per inhabitant)	2006–2018	T
Social benefits by disability (purchasing power standard (PPS) per inhabitant)	2007–2018	T
Social benefits by family/children (purchasing power standard (PPS) per inhabitant)	2007–2018	T
Social benefits by sickness/health care (purchasing power standard (PPS) per inhabitant)	2007–2018	T

Note: Most factors were variables collected for men and women, separately. However, in some cases, the poor availability of absenteeism characteristics led us to use the “total” information in the estimation process.

**Table 2 ijerph-18-11823-t002:** Statistics of the absence rate due to own illness or disability for European men and women (average, 2006–2020).

Mean		Median	CV	Min	Max	MW
Men	2.8%	1.2%	158.9%	0.1%	20.4%	375 **
Women	3.7%	1.5%	151.3%	0.1%	22.9%	

Note: *n* = 32 countries. CV is the coefficient of variation; significance levels: ** *p* ≤ 0.05; MW—Mann-Whitney U test [64]. We used nonparametric statistics because the test of normality carried out (here the Shapiro–Wilk test) rejected the null hypothesis of normality. The MW test does not assume a normal distribution of the variables, unlike the analogous one-way analysis of variance and Student’s *t*-tests [65].

**Table 3 ijerph-18-11823-t003:** Diagnostics of the modeling for men’s and women’s absence rate from work due to own illness or disability.

Diagnostics	Men		Women	
	OLS	GWR	OLS	GWR
R-Squared	0.64	0.71	0.56	0.77
Adjusted R-Squared	0.58	0.87	0.48	0.75
AICc	141.4	103.5	107.1	58.7
Moran’s I	0.06 **	0.008	0.12 **	0.005
Joint Wald Statistics	113.4 ***	-	59.6 ***	-
Koenker (BP)	21.80 *	-	35.62 **	-
Jarque–Bera	5.6 *	-	1.1	-

Note: the Moran’s I statistic showed spatial autocorrelation in the OLS residuals (initially, it was confirmed by the ESDA results, Table A1 and Figure A1). Moreover, the Koenker (BP) statistic indicated that the OLS modeled relationships were not consistent either, due to non-stationarity or heteroscedasticity. Significance levels: *** *p* ≤ 0.01, ** *p* ≤ 0.05, * *p* ≤ 0.1.

**Table 4 ijerph-18-11823-t004:** Local values of the GWR coefficients for men’s and women’s absence rate from work due to own illness or disability.

	Men					Women				
	BMI	ARSJ	MEM	SBSHC	PA	PRP	LSI	WEM	SPB	CDP
AT	7.2	−9.9	3.5 ***	2.6 *	−1.4 *	2.4	1.3	2.1 *	2.7 *	0.9
BE	8.4	−17.5	3.6 **	1.6	−0.5	2.3	2.9	1.9	2.2	2.1 *
BG	5.6 **	−4.1	3.1 ***	1.9 **	−0.9 *	2.9	1.6 *	2.5 **	3.3 ***	1.1
CH	7.3 *	−12.7	3.6 ***	2.5 *	−1.1	3.6	2.1	2.1 *	2.9 **	1.6
CY	6.1 **	−4.5	3.1 ***	1.9 ***	−0.8 *	3.2 *	1.7 *	2.6 ***	3.6 ***	1.2 *
CZ	8.8	−10.1	3.6 **	2.4 *	−1.7 *	2.2	2.4	1.8	2.2	1.4
DE	9.1	−14.1	3.7 ***	1.9	−0.9	2.6	2.8	1.9	2.4	2.1 *
DK	3.9	−12.5 *	2.8 ***	2.1 **	−0.4	2.9	2.6 **	2.1 **	3.1 **	1.4 **
EE	−4.8	−5.8	0.7	1.8	−0.1	0.8	0.8	1.3	2.5	0.8
ES	6.4 **	−12.6 **	3.1 ***	2.2 **	−0.6	3.1 *	1.8 *	1.8 **	3.1 ***	0.7
FI	−0.9	−7.9	1.3	1.9 *	−0.1	1.9	1.2	1.6	3.2 **	0.6
FR	6.9 *	−16.5 *	3.2 ***	2.1 *	−0.7	3.1 *	2.2 *	1.9 *	2.7 **	1.1
GB	6.5	−23.1 *	2.9 **	1.5	−0.1	3.2	3.2 **	1.9 *	2.8 *	1.6 *
GR	6.2 **	−4.8	3.2 ***	1.9 **	−0.9 *	3.4 *	1.7 *	2.7 ***	3.7 ***	1.1
HR	5.5	−6.1	2.9 **	2.3 **	−1.2 *	1.4	0.2	2.1	2.7	0.1
HU	4.8	−3.3	2.5 *	1.9	−1.1	1.3	1.3	1.1	1.7	0.7
IE	6.2 *	−18.3 **	2.9 ***	2.1 **	−0.4	3.9 **	2.5 **	2.1 **	3.3 ***	1.2 **
IS	6.7 **	−13.1 **	2.9 ***	2.3 ***	−0.6	4.9 ***	2.5 ***	2.5 ***	4.3 ***	1.1**
IT	6.5 **	−8.0	3.3 ***	2.4 **	−1.1 *	3.7 *	1.7	2.4 **	3.6 ***	1.0
LV	−2.4	−3.4	1.3	1.8 *	−0.3	1.2	1.1	1.6	2.6 *	0.9
LT	0.4	−3.3	1.9 **	1.9 **	−0.5	1.7	1.4	1.8 *	2.8 **	1.4 *
LU	9.5	−14.4	3.9 **	1.9	−0.5	1.8	2.9	2.0	2.3	2.7
MT	6.8 ***	−6.7	3.2 ***	2.2 ***	−0.9 *	3.6 **	1.8 **	2.5 ***	3.8 ***	1.1*
NL	6.6	−16.1 **	3.3 ***	1.9 **	−0.6	2.5	2.8 *	1.9 *	2.5	1.8 **
NO	2.6	−9.9	2.3 **	2.1 ***	−0.3	3.2 *	2.1 *	2.2 **	3.8 ***	0.8
PL	2.8	−4.5	2.5 **	1.9 ***	−0.9 *	2.6	1.9	2.1 *	2.8 **	1.7 *
PT	6.3 ***	−11.9 ***	3.1 ***	2.2 ***	−0.6	3.2 **	1.8 **	1.9 **	3.2 ***	0.7
RO	5.1 *	−3.3	2.9 ***	1.8 **	−0.9 *	2.6	1.6 *	2.2 **	2.9 **	1.3
SE	−1.2	−8.7	1.6	1.9 *	0.1	1.9	1.7	1.7	3.1	0.8
SI	6.1	−7.5	3.2 **	2.4 *	−1.3 *	1.7	0.5	1.9	2.6	0.3
SK	4.7	−3.4	2.8 **	1.9 **	−1.1 *	1.9	1.8	1.5	2.1	1.2
TR	5.7 **	−3.9	3.1 ***	1.9 **	−0.9 *	3.1 *	1.7 *	2.6 **	3.5 **	1.3

Note: Significance levels: *** *p* ≤ 0.01, ** *p* ≤ 0.05, * *p* ≤ 0.1.

## Data Availability

The data presented in this study are openly available from Eurostat, the WHO, and the OECD websites.

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
