# Peer review of "Causes of Sickness Absenteeism in Europe—Analysis from an Intercountry and Gender Perspective"

_ijerph, 2021, doi:10.3390/ijerph182211823_

Round 1
Reviewer 1 Report
a). The title mentions “causes of absenteeism”, but does not indicate what type of absenteeism it is. This is understood when the document is read, however, it must be considered that the title is itself one of the main metadata.
b). The reasons for studying the causes of absenteeism are not understood, it seems only a descriptive analysis during a period of time, by country and by gender, derived from a private or government database, but without an apparent purpose.
c). The keywords are incorrect, they are not actually keywords, many are broad phrases. It is recommended to use a Thesaurus.
d). The document in general requires a strict revision in its writing. It is made up of excessively long paragraphs where multiple ideas are included in one. You must use “full stop”.
e). Separate introduction and theoretical framework, there is a confusion of sections that constitutes a scientific article.
f). No research objectives (general and specific) are identified, or research questions, hypotheses or whatever applies to this type of study. Since these kinds of elements are not included, the purpose of the study and the relationship that exists in studying countries and gender (which I think is actually sex because it considers male and female) is unknown.
g). In the methodology section, it should not include any data analysis, but rather define the sample, type of sampling, means of data collection, type of research, research focus, etc., etc.
h). Figure 2 does not present a logical order in the data included, it is assumed that the order should be ascending or descending and not by country in general, even without alphabetical order. It should also be considered that it includes too many variables not only of sex but also a large number of countries, which should be grouped for the analysis in blocks. A graph with more than 12 variables is no longer functional for the reader or for its correct interpretation.
i). The Exploratory Data Analysis section does not go into materials and method.
j). It seems that the discussion takes more relevance than the data analysis itself and it should be the opposite.
k). Avoid quotes in the conclusions.
l). The information provided in the Data Availability Statement should appear as a more specific explanation of the method section.
m). It is necessary to define the focus of the study regarding its area of ​​application: health, economics, administration, sociology, etc.
n). The document must observe the recommended structure for a research article: introduction, theoretical framework, objectives, methodology, data analysis, discussion / conclusions, references, annexes.
Author Response
Dear Reviewer,
we corrected the paper according to all remarks. Comments were highly insightful and enabled us to greatly improve the quality of the manuscript. Please, find below point-by-point responses to each of the comments.
Comments to the Author:
a). The title mentions “causes of absenteeism”, but does not indicate what type of absenteeism it is. This is understood when the document is read, however, it must be considered that the title is itself one of the main metadata.
Thank you very much for this remark. We have changed title into: Causes of Sickness Absenteeism in Europe – Analysis from an Intercountry and Gender Perspective
b). The reasons for studying the causes of absenteeism are not understood, it seems only a descriptive analysis during a period of time, by country and by gender, derived from a private or government database, but without an apparent purpose.
We agree with this suggestion. We have added two paragraphs on the motivation of modelling on absenteeism causes. One is in the end of the Introduction and the second is the new section on Literature review:
“As previously mentioned, planned, desired absences from work (e.g., annual holidays) can easily be “absorbed” by companies, as their effects can be planned for and mitigated; however, unplanned, undesired absences (e.g., illnesses, lay-offs) can disrupt production cycles and lead to material losses for both employers and employees. Lost productivity, labor turnover costs, and the disability burden have made sickness absence one of the top policy matters throughout Europe. [22]. Although there is large number of evidence on the different factors associated with sickness absenteeism on transnational European trends are far more scarce. Although there is a large amount of evidence available on the different factors associated with sickness absenteeism, studies on transnational European trends are far more limited. International comparisons are thus urgently needed, since they may not only help to assess a country’s economic performance, but also enable overall patterns across countries to be observed, thereby indicating which policies are working. This could be valuable for both public health and economic reasons. Additionally, research and intervention priorities can be identified from the examination of similarities and differences in sickness absence between countries. Finally, this work could indicate countries where important steps should be taken to help reduce sickness absence across the Europe. As mentioned before, sickness absence is the basis of an employee’s absence from work. It is a problem not only from the employer’s perspective, but also from the perspective of the state and lost GDP [22]. Thus, sickness absence, often called sickness absenteeism, is most often understood as the number of missed working days by employees due to diseases. It is also defined as the absence from work that is attributed to an employee’s sickness and accepted as such by the employer [23]. We try, therefore, to fill the knowledge gap concerning disability, work absenteeism, and sickness benefits in selected European countries. This study thus aims to extract the transnational causes of sickness absenteeism in men and women separately. This approach was associated with significant differentiation in the value of the absence rate in terms of gender, which was confirmed in the literature on the subject [23–25]. Moreover, the following research questions have been posed:
- Is the disease spectrum different in the studied countries?
- Why and where are the sickness absence percentages higher for men than for women?
- Do social policy and sick-leave legislation systems influence sick-ness absence trends?
- Do work-related factors, such as job satisfaction, affect sickness absenteeism?”
And:
“To the best of our knowledge a study that considers such a broad approach has never been carried out so far. The results of this study but should be therefore relevant when to formulating strategic labor and healthcare policy recommendations. They could provide information for European governments, employers and policy makers to de-sign strategies that aid to reduce economic expenditures, increase workers satisfaction, assist employers, and eventually facilitate economic development of the country. Moreover, our findings suggest establishing changes in the European sickness absence legislation to make available comparable official data between countries. Finally, the data provide preliminary evidence where to look for practices which can be implemented.”
c). The keywords are incorrect, they are not actually keywords, many are broad phrases. It is recommended to use a Thesaurus.
Thank you very much for this suggestion. We have changed the Keywords into short phrases: “Keywords: sickness absenteeism; gender inequalities; socioeconomic factors; Europe; regionality; geographically weighted regression”
d). The document in general requires a strict revision in its writing. It is made up of excessively long paragraphs where multiple ideas are included in one. You must use “full stop”.
We agree with this suggestion. We have separated form and Introduction a new section on literature revision. We have also shortened the Materials and methods paragraph (a huge part of the data analysis we put into the Appendix as a robustness check of the results). We have also shortened the discussion section and corrected Conclusions. The text was revised by the English Speaker Editor.
e). Separate introduction and theoretical framework, there is a confusion of sections that constitutes a scientific article.
We would like to thank the Reviewer for these constructive remarks. We have divided the Introduction into Introduction itself and the second section: 2. Theoretical framework on the sickness absence determinants.
f). No research objectives (general and specific) are identified, or research questions, hypotheses or whatever applies to this type of study. Since these kinds of elements are not included, the purpose of the study and the relationship that exists in studying countries and gender (which I think is actually sex because it considers male and female) is unknown.
In this point we fully accept the Reviewer’s opinion. We have added a separate paragraph on objectives. Please find it in the Introduction.
“We try, therefore, to fill the knowledge gap concerning disability, work absenteeism, and sickness benefits in selected European countries. This study thus aims to extract the transnational causes of sickness absenteeism in men and women separately. This approach was associated with significant differentiation in the value of the absence rate in terms of gender, which was confirmed in the literature on the subject [23–25]. Moreover, the following research questions have been posed:
- Is the disease spectrum different in the studied countries?
- Why and where are the sickness absence percentages higher for men than for women?
- Do social policy and sick-leave legislation systems influence sick-ness absence trends?
- Do work-related factors, such as job satisfaction, affect sickness absenteeism?”
g). In the methodology section, it should not include any data analysis, but rather define the sample, type of sampling, means of data collection, type of research, research focus, etc., etc.
l). The information provided in the Data Availability Statement should appear as a more specific explanation of the method section.
Thank You for these remarks (g) and (l). We removed the data analysis section from the Materials and methodology – we have moved it into the Results. At the same time, we have added the note about the data (the source, sample and availability) in the Materials.
h). Figure 2 does not present a logical order in the data included, it is assumed that the order should be ascending or descending and not by country in general, even without alphabetical order. It should also be considered that it includes too many variables not only of sex but also a large number of countries, which should be grouped for the analysis in blocks. A graph with more than 12 variables is no longer functional for the reader or for its correct interpretation.
We have changed and clarified the Figure 2. Thank You.
i). The Exploratory Data Analysis section does not go into materials and method.
In accordance to this Reviewer’s remark, we have removed the ESDA analysis from the Materials and method and out it into Appendix. We considered this as a kind of robustness check of the data analysis outcomes. We believe that now it sounds more relevant. Thank You.
j). It seems that the discussion takes more relevance than the data analysis itself and it should be the opposite.
We would like to thank the Reviewer for these constructive remarks. We have removed some data from the Discussion, shortened this paragraph and transformed therefore the Data analysis section.
k). Avoid quotes in the conclusions.
We have removed the quotes from the Conclusions.
m). It is necessary to define the focus of the study regarding its area of ​​application: health, economics, administration, sociology, etc.
We agree with this suggestion. We have added some policy recommendation (in the Introduction, in the end of the second paragraph and in the Conclusions) and therefore we highlighted has some practical and applicative potential benefits of our study.
n). The document must observe the recommended structure for a research article: introduction, theoretical framework, objectives, methodology, data analysis, discussion / conclusions, references, annexes.
We appreciate this remark very much. The whole body of the paper was revised and transformed. The study was also proofread by the Native Speaker Editor. We attached the Certificate to the review section.

Reviewer 2 Report
The article deals with an interesting topic like studying the absenteeism in Europe from a cross-country level of analysis. Including the years of the pandemic could affect as a bias effect some of the information and results and I believe that this must be taken into account as a limitation in the final discussion. From my point of view the article follows an academic structure, presenting rigor in the literature review (most of the relevant literature are covered), methods and analysis. The article provides a rich discussion at the end with the main values extracted from the analysis.
Author Response
Dear Reviewer,
we appreciate Your decision, and we would like to thank You for this comment.
Comment to the Author
The article deals with an interesting topic like studying the absenteeism in Europe from a cross-country level of analysis. Including the years of the pandemic could affect as a bias effect some of the information and results and I believe that this must be taken into account as a limitation in the final discussion. From my point of view the article follows an academic structure, presenting rigor in the literature review (most of the relevant literature are covered), methods and analysis. The article provides a rich discussion at the end with the main values extracted from the analysis.
Thank you very much for this recommendation.
Reviewer 3 Report
This paper purposed to explore the causes of absenteeism in Europe – analysis from a cross country and gender perspective. I do have some comments as listed below in the order noted.
Comment 1: The Abstract looks like the Introduction of the background of the study. Please revise it.
Comment 2: Do the Austria (AT) and Turkey (TR) are one of the 32 European countries?
Comment 3: Please provide the significance level P values in Figure 2.
Author Response
Dear Reviewer,
we appreciate Your decision, and we would like to thank you for the comments. We corrected the paper according to the remarks and clarified the suggestions. Please, find below point-by-point responses to each of the comments.
Comments to the Author
Comment 1: The Abstract looks like the Introduction of the background of the study. Please revise it.
Thank you for this remark. We have rewritten the abstract taking into consideration the Reviewer’s suggestion.
Comment 2: Do the Austria (AT) and Turkey (TR) are one of the 32 European countries?
Our explanation to Austria and Turkey.
Austria is a country of south-central Europe and is bordered to the north by the Czech Republic, to the northeast by Slovakia, to the east by Hungary, to the south by Slovenia, to the southwest by Italy, to the west by Switzerland and Liechtenstein, and to the northwest by Germany. It extends roughly 360 miles (580 km) from east to west. Austria is an EU member country since 1 January 1995 (https://europa.eu/european-union/about-eu/countries/member-countries/austria_en, accessed on 5 November 2021).
Turkey is a country that occupies a unique geographic position, lying partly in Asia and partly in Europe. Throughout its history, it has acted as both a barrier and a bridge between the two continents. However, in 1987, Turkey applied to join what was then the European Economic Community, and in 1999 it was declared eligible to join the EU. Turkey's involvement with European integration dates back to 1959 and includes the Ankara Association Agreement (1963) for the progressive establishment of a Customs Union (ultimately set up in 1995). Accession negotiations started in 2005, but until Turkey agrees to apply the Additional Protocol of the Ankara Association Agreement to Cyprus, eight negotiation chapters will not be opened, and no chapter will be provisionally closed. Finally, Turkey is a key strategic partner of the EU on issues such as migration, security, counter-terrorism, and the economy (https://ec.europa.eu/neighbourhood-enlargement/enlargement-policy/negotiations-status/turkey_en, accessed on 5 November 2021).
Comment 3: Please provide the significance level P values in Figure 2.
Thank you very much for this suggestion. You can find the significance levels under Figure 2 in Note.
Round 2
Reviewer 1 Report
All very good, congrats.